# Alpine Musk Deer (*Moschus chrysogaster*) Adjusts to a Human-Dominated Semi-Arid Mountain Ecosystem

**DOI:** 10.3390/ani12213061

**Published:** 2022-11-07

**Authors:** Lixun Zhang, Zhangyun Sun, Bei An, Dexi Zhang, Liuyang Chen

**Affiliations:** 1College of Ecology, Lanzhou University, No. 222, Tianshui South Road, Lanzhou 730000, China; 2Yuzhong Mountain Ecosystems Observation and Research Station, Lanzhou University, Lanzhou 730000, China; 3School of Basic Medicine Sciences, Lanzhou University, Lanzhou 730000, China

**Keywords:** camera traps, Alpine musk deer, activity rhythm, temporal-spatial responses, human-dominated

## Abstract

**Simple Summary:**

Long-term monitoring of the Alpine musk deer (AMD, *Moschus chrysogaster*) population is essential in any human-dominated landscape as their population is globally showing a declining trend due to high deforestation rates, hunting pressure, and human disturbance. In view of that, the AMD population was monitored in Xinglong Mountain Nature Reserve (XMNNR), as their past high-density distribution areas, northwest of China, using a camera trap between September 2018 and August 2020. The results showed that AMD strongly spatial temporal avoids livestock but gradually adjusts to human activities. Compared to the previous report, AMD distributes artificial forests that are not confined to the areas reported but also cover other potential areas.

**Abstract:**

Comprehension of whether human and livestock presence affects wildlife activity is a prerequisite for the planning and management of humans and livestock in protected areas. Xinglong Mountain Nature Reserve (XMNNR) in northwest China, as a green island in a semi-arid mountain ecosystem, is one of the scattered and isolated areas for Alpine musk deer (AMD), an endangered species. AMD cohabits their latent habitat area with foraging livestock and humans. Hence, habitat management within and outside the distribution areas is crucial for the effective conservation of AMD. We applied camera traps to a dataset of 2 years (September 2018–August 2020) to explore seasonal activity patterns and habitat use and assess the impacts of AMD habits in XMNNR. We investigated AMD responses to livestock grazing and human activities and provided effective strategies for AMD conservation. We applied MaxENT modeling to predict the distribution size under current conditions. The activity patterns of the AMD vary among seasons. The optimum habitat average distance to cultivated land ranges of AMD (150~3300 m during grass period/100~3200 m during withered grass period), distances to the residential area ranges (500~5700 m during the grass period/1000~5300 m during the withered grass period), elevation ranges (2350~3400 m during the grass period/2360~3170 m during the withered grass period), aspect ranges (0~50° and 270~360°), normalized vegetation index ranges (0.64~0.72 during the grass period/0.14~0.60 during the withered grass period), and land cover types (forest, shrub, and grassland). Results present that the predicted distributions of AMD were not confined to the areas reported but also covered other potential areas. The results provide evidence of strong spatial-temporal avoidance of AMD in livestock, but gradually adjusting to human activities. These camera trap datasets may open new opportunities for species conservation in much wider tracts, such as human-dominated landscapes, and may offer guidance and mitigate impacts from livestock, as well as increase artificial forest planting and strengthen the investigation of the potential population resources of AMD.

## 1. Introduction

Comprehension of whether human and livestock presence affects wildlife activity is a prerequisite for conservation planning in protected areas, especially in human-dominated landscapes [1,2]. Human disturbance and livestock grazing may impose disturbance on wildlife survival by altering habitat quality [3]. Wildlife may consider humans or livestock as potential threats and, subsequently, move away in space or time to avoid humans and livestock [4]. Avoidance could present spatial partitioning, in which animals preferentially occupy areas with lower human activity, or temporal partitioning, such that humans and wildlife are sympatrically distributed at different times [5]. Local, ongoing, high-resolution monitoring of human-wildlife interaction may facilitate more realistic and sufficient incorporation of the experienced impacts of human-wildlife conflict in conservation planning and management [6]. Long-term monitoring of endangered species is vital in any human-dominated area as their population is globally decreasing [7].

Alpine musk deer (*Moschus chrysogaster*, Hodgson, 1839; hereafter AMD) is a solitary endemic species of the Qinghai-Tibetan Plateau, mainly scattered and isolated in high-altitude coniferous forests or broad-leaved forests, with Alpine shrub meadows [8]. AMD is classified as an endangered species according to the IUCN Red List criteria (IUCN, 2016) due to habitat loss and degradation [8]. Reforestation is one of the most productive approaches to coping with climate warming [9]. Large-scale afforestation has been conducted by the Chinese government for the past three decades [10]. Forest restoration is also a key means to reversing the ecosystem degradation caused by human activities worldwide [10]. Due to climate change, AMD should migrate to the northeast (to higher latitudes) [11]. However, a species along with climate warming will be doomed to extinction due to inaccessible climatically suitable habitats because of geographic barriers, or the species’ inability to disperse [12]. In situ protection is the best option for their protection measures [13] due to their isolated distribution at the highest altitude. Hence, habitat management within and outside protected areas is crucial for the effective conservation of AMD. The survival of AMD depends on the surrounding environmental conditions. Ecological factors (e.g., vegetation types, altitude, slope) in combination with human disturbance, which have important influences on the distribution of AMD [13], are usually applied to predict their distributions.

Considerable information is available on AMD’s habitat degradation [14] and suitability assessment [12,13], population distribution, and influencing factors [15,16]. With the ecotourism industry blooming around the globe, investigation of the behavioral effects of a human visit to preservation areas, especially in endangered species distribution areas, should be a conservation priority [17]. Most AMD studies so far have used indirect accounts such as pellet counts and interviews to conduct musk deer distributions [16], or conflict issues [3,15], or identified publications [14] as an index of AMD abundance in the field. A major challenge is to monitor the AMD populations over the period, as they are elusive in the mountains or forests, making their observation challenging. Furthermore, forests or shrubs can hamper animal detectability, as animals hidden behind shrubs, and forest-dwelling mountain ungulates may have activity peaks at night [18] and are rarely observed during daytime surveys [16]. Fewer animal encounters during traditional survey techniques in disturbed areas may lead to inaccurate population estimation compared to undisturbed areas [19].

Camera traps are sufficient devices for spotting elusive and rare solitary mountain ungulates in remote habitats, improving abundance estimations [20], and have been extensively applied to survey activity patterns of wildlife species without observer interference or disturbance [21]. Therefore, we applied camera traps to investigate the spatial and temporal responses of AMD to livestock and human interference.

There is no capability to carry out a complete investigation of a target species on informative temporal-spatial scales, and data on species distribution sites is erratic [22]. Hence, reliable field estimates should be conducted, based on which a scientific management plan should be designed and then be implemented [14]. The MaxENT model is the most widely used for the simulation and assessment of wildlife distributions [23] and predicts the suitable geographical habitats of species and outperforms other models due to good performance with inadequate dataset requirements, less model running time, friendly operation, small sample size prerequisites, and great simulation precision [12,24,25,26]. Several musk deer studies on current and future climatic-niche predictions using MaxENT have been carried out to identify these biogeographic signals and verify the model [12,13,25,26].

As a green island in a semi-arid mountain ecosystem, Xinglong Mountain Nature Reserve (hereafter XMNNR, 35°38′–35°58′ N; 103°50′–104°10′ E, 29, 583 ha, Figure 1) in northwest China, is one of the scattered and isolated areas for AMD. The area receives 625 mm of annual precipitation, which can satisfy AMD’s annual precipitation needs [11]. The population densities of AMD in XMNNR declined due to illegal hunting and habitat destruction from deforestation. There is a wide gap in protected natural areas in XMNNR, which are highly exploited by humans. Additionally, its landscape is intermixed with human disturbances such as medicinal plants, ecotourism, and livestock grazing pervasively across the landscape, including protected areas [16]. Wildlife has been protected by the Chinese government and artificial forests have been planted since the implementation of China’s Three-North Shelterbelt Forest program in the 1980s, soil fertility and water conservation function have been maintained [27]. There is no predatory pressure in XMNNR due to the disappearance of carnivorous animals since 1960. Hence, we investigated the temporal-spatial distribution and seasonal activity patterns of AMD by line transects and camera-trap and clarified the main factors likely to affect AMD’s suitable habitats in XMNNR to estimate AMD’s suitable habitat areas by the MaxENT model.

## 2. Materials and Methods

### 2.1. Camera Trap Survey

Five monitoring areas (GTG, MJS, XLS, MXS, and SZ GTG—Guantangou, MJS— Majiasi, XLS—Xinglongshan, MXS—Maxianshan, and SZ—Shangzhuang) are divided into 1 km^2^ grid cells by the Geographic Information System (ArcGIS 10.2). Sixty camera traps (LTL 6210, EAGLE E1B, and Seagull-LY-1, Shanghai, China) were deployed at altitudes of 2000–3600 m from September 2018 to August 2020 (Figure 1). We selected open and front lighting to deploy the camera trap in each grid cell. Camera traps were mounted 0.3–1.0 m above the ground with small branches or bryophytes and were set to operate 24 h per day by taking 3 photographs and 30 s of video with a delay of 60 s between consecutive exposures. We recycled photographs and videos and recharged the battery every 3–4 months.

### 2.2. Potential Distribution Assessment

#### 2.2.1. Environmental Variables Obtained and Preprocessing

The MaxENT model [24] was applied to assess the suitable habitat of AMD species occurrence data and environmental variables to predict their potential habitat in large-scale spaces. We scored the following variables: (i) topographic variable data: altitude, slope, and aspect are extracted from the Digital Elevation Model (DEM) with a resolution of 30 m from the scientific database of the Chinese Academy of Sciences through ArcGIS raster analysis tools [28]; (ii) vegetation variable data: land cover type from global data provided by the Chinese government and normalized difference vegetation index (NDVI) [29]; and (iii) human disturbance variable data: distance from cultivated land, distance from the road, and distance from the residential area, which were calculated from the Euclidean distance in ArcGIS 10.2 using the vectors of rivers, roads, and residential points, respectively. All environmental variables were resampled, and the raster layer with a uniform resolution of 1 km was converted into “.asc” format by ArcGIS 10.2, which is required by the MaxENT software [24]. The species distribution coordinate point data obtained by the camera trap and line transact are screened for spatial autocorrelation (about 0.3 × 0.3 km) to avoid over-fitting the spatial distribution using the “Wallace” package (https: //wallaceecomod.github.io/, accessed on 19 May 2021) [30] based on each grid cell having a maximum of one occurrence point. Eighty (green grass period) and sixty-five (withered grass period) distribution points of AMD were applied for their habitat suitability prediction.

#### 2.2.2. Model Procedure

MaxENT 3.4.1 (http://www.cs.princeton.edu/~schapire/maxent/ accessed on 22 May 2021) [24] based on presence-only data [31] was conducted to predict the potential habitat suitability of AMD. We used 10 bootstrap replicates and randomly split the presence records into training and testing data (75 and 25%, respectively). A pair of variables with Pearson’s correlation coefficient > |0.80| was removed and remained the most important factor for consideration in the final model according to permutation importance and ecological meaning for AMD [32]. Jackknife and logistic output were selected to comprehensively assess the impact of environmental factors and represent the logistic values ranging from 0 (lowest probability) to 1 (highest probability) for the distribution of potentially suitable habitat.

The area under the curve (AUC) of receiver operating characteristics [33] was applied to model performance. A larger AUC value (range 0~1) indicates a better prediction effect of the model. We assumed that 10% of the distribution points were spatially biased and used the trained logical value to determine the threshold. This threshold was applied to classify habitats as suitable (≥threshold value) or unsuitable (<threshold value) for AMD.

### 2.3. Data Analyses

To understand their spatial uses, we collected occurrence data of AMD, humans, and livestock using a camera trap in two periods based on their food phenological characteristics (green grass period, March–September, and withered grass period, October–February) [34]. To evaluate their abundance in different study areas, we used the relative abundance index as a substitute for relative abundance; that is, the number of ICs of the same species taken in each area multiplied by 1000 and then divided by the total camera-days (24 h was defined as one camera-day) of the area [35]. The daily activity pattern was analyzed by the kernel density method [36]. To explore the degree of interspecific overlap in activity between AMD and humans or AMD and livestock across two periods, we estimated the diel activity for three species by fitting a non-parametric circular kernel density estimation function [36] to the radian-transformed occurrence points, considered as a random sampling from 24 h per day, presenting the animals’ maximum true diel activity pattern [36]. We conducted and visualized using the “overlap” package [37] and applied the coefficient of overlap (∆_4_ or ∆_1_) to determine temporal niche differentiation between three species, whereby ∆ can range from 0 (no overlap) to 1 (complete overlap) [36]. When the small samples (detection records) were less than 75, ∆_4_ was applied as the parameter estimation; otherwise, ∆_1_ was used [37]. Then, we computed the 95% confidence intervals (hereafter, 95% CIs) for more robust overlap estimates by using 10,000 bootstrap replicates [38]. To test whether there were differences in the kernel density curves between these three species across seasons, the Wald test with the “activity” package [39] was used to obtain the test value (significance level at 0.05) by nonparametric bootstrap resampling with 10,000 iterations [38]. All analyses were performed using R version 4.0.5 [40].

## 3. Results

### 3.1. Spatial Distribution of AMD, Livestock Grazing, and Human Activities

Four hundred and sixty-one AMD independent captures (from 45 sites), 612 independent captures of human activity (from 39 sites), and 1019 independent captures of livestock grazing (from 25 sites) were collected from 30713 camera trap days by 60 camera traps. Of the 60 camera trap sites monitored (Figure 2), 40% (24/60) were used by both AMD and human activities (Figure 2). Only 30% (18/60) were used by both AMD and livestock.

The elevational distribution of AMD (2775.49 ± 170.00 m) in the green grass period was significantly higher than that (2690.49 ± 185.59 m) during the withered grass period (Z = −4.45, *p* < 0.01). The elevational distribution of AMD (2775.49 ± 170.00 m) was significantly lower than that of grazing (3186.29 ± 357.35 m, Z = −18.52, *p* < 0.01). The elevational distribution of AMD (2775.49 ± 170.00 m) was significantly lower than that of grazing (3186.29 ± 357.35 m, Z = −4.45, *p* < 0.01) and significantly higher than that of human activity (2442.71 ± 134.78 m, Z = −20.72, *p* < 0.01) in the green grass period (Figure 3). The elevational distribution of AMD (2690.97 ± 185.59 m) was significantly lower than that of livestock grazing (3146.08 ± 291.77 m; Z = −8.72, *p* < 0.01) and significantly higher than that of human activities (2455.83 ± 129.66 m; Z = −8.44, *p* < 0.01) during the withered grass period (Figure 3).

### 3.2. Temporal Activity Patterns

The temporal activity pattern of AMD differed during the green grass period from the withered grass period. The daily activity pattern of AMD showed a bimodal pattern (9:00~11:00 and 19:30~20:30). In the withered grass period, the daily activity pattern of AMD showed three peaks (11:00~13:00, 19:30~20:30, and 2:00~4:00). Temporal overlap analysis of AMD showed a considerable overlap (Δ_4_ ≥ 0.5) with livestock but less overlap (Δ_4_ ≤ 0.5) with humans during both seasons (Figure 4).

### 3.3. MaxENT Model Prediction Results and Habitat Suitability Evaluation

The average test AUC value for the model was 0.90 during the green grass period and 0.94 during the withered grass period, respectively (Appendix A), indicating that it predicted AMD habitat very well with high levels of accuracy. Model predictions matched the collected occurrences of AMD and showed a potential geographic range in XMNNR (Figure 5). The current predicted suitable habitat of AMD is approximately 11,183.27 ha during the green grass period (Figure 5). The suitable habitat of the AMD is mainly distributed in grasslands and forests. The suitable habitat area of AMD in XMNNR was 8599.03 ha during the withered grass period. Compared with the green grass period, suitable habitats for forests and grasslands have decreased during the withered grass period, while suitable habitats for cultivated land and shrubs have increased.

### 3.4. Habitat Selection Preference

The potential suitable distribution of the *AMD* was distance to cultivated land, NDVI, land coverage type, and distance to the road during both the green grass period and withered grass period (Appendix A), but NDVI is more important during the green grass period (Appendix A). We also found that factors of aspect, distance to rivers, and roads contributed little to habitat suitability for both species (Appendix A). Response curves revealed the direction of effects by the five most important variables in the model for the suitable distribution for Alpine musk deer during the green grass period and withered grass period (Appendix A).

Distance to cultivated land and distance to a water source were the best predictors of AMD habitat distribution, contributing 27.2% and 27.0% during the green grass period, respectively, and 30.4% and 29.9% during the withered grass period, respectively (Appendix A).

AMD was primarily found in shrubs and forests (Appendix A). The probability of occurrence decreased sharply as the aspect increased (>20°), as shown in Appendix A. We found that the probability of occurrence of AMD showed an obvious increase with the value of NDVI greater than 0.4 during the green grass period (Appendix A) but 0.5 during the withered grass period (Appendix A). AMD mainly distributes mid-elevation ranges (2600–3400 m a.s.l, Appendix A) during the withered grass period, but slightly lower at high elevation ranges (2600–3000 m a.s.l, Appendix A) during the withered grass period.

## 4. Discussion

Society has an ethical responsibility to pursue an increasingly sustainable use of natural resources. In comparison with the results of surveys conducted earlier [15], the current survey data presents a wider distribution in the numbers of AMD in the reserve and especially in the adjacent territories. This study is the longest two-year monitoring period and is more objective and reliable than previous documents [16]. It supports the notion that long-term monitoring of endangered species is vital in any human-dominated landscape, as their population is showing a declining trend worldwide [7]. Forty percent of camera trap sites were occupied by both AMD and human activities, supporting the conclusion that animals and humans can coexist at fine spatial scales [1,21] while only 30% were shared by both AMD and livestock. Hence, we found strong proof AMD spatially avoided livestock in the reserve and adjacent territories, which confirmed the negative influences of livestock [14]. Due to their being more specialized feeders than livestock, AMD used 17% exclusively [26].

The altitude of AMD in the green grass period was significantly higher than that during the withered grass period (Figure 3), which mirrors previous findings of AMD seasonal migration by camera trap data [13]. The present study indicated a high habitat overlap between AMD and humans or livestock (Figure 2) but elevational avoidance during both seasons (Figure 4). In this study, AMD was most active during the early morning and late evening during the green grass period. A considerable amount of temporal overlap was found between livestock and associated activities with AMD, largely contributed by activity overlap during the morning hours (6~9 a.m.). AMD showed a higher peak in the late evening hours, which could be a strategy of competitiveness in space, in which species of domestic animals appear during the afternoon. Hence, the coexisting animal species segregate primarily by their habitat’s specializations [15]. Another possible explanation is that it might be due to a lower presence of humans and livestock because seasonal changes in temperature and food are expected to influence the activity patterns of the animals [41].

Distance to cultivated land and distance to a water source were the strongest predictors of AMD habitat distribution during the green and withered grass periods (Appendix A). This partly supports the hypothesis that changing food, concealment, and water sources would influence the space use of the AMD [12,14] as well as other mammals [41]. We have found distance to cultivated land affected AMD habitat selection, which has been confirmed that musk deer strongly avoided agricultural areas [16]. At present, AMD has the highest utilization rate of shrubs, with forests and grasslands having the second lowest utilization rates of arable land during both green grass and withered grass periods. Nevertheless, nearly 70% of AMD is in shrubs, and there is no wild AMD in artificial forest habitat [16]. This disagreement might be caused by logistical constraints associated with rugged terrain and harsh weather that affect traditional methods. Our results echo early studies showing similar phenomena in artificial forest habitats, which may be a solution for AMD as reforestation is one of the most productive approaches for coping with climate warming [9] and provides a promising approach for protection in situ, especially for species with a narrow distribution.

Notably, human activities occur mostly in AMD-suitable areas in XMNNR (Figure 2), but AMD avoids them in elevational (Figure 3) and temporal dimensions (Figure 4). The general pattern of habitat utilization and selection of AMD is an adaptive strategy to human activities due to long-time coexistence at fine spatial scales [3] within small human-dominated isolated island areas [12]. The suitable habitat of the AMD was more densely distributed in XLS and MJS with a continuous area (Figure 5), in which dense forest habitats provide feedback with more resources and refuge from predation [15,16]. Notably, based on the MaxENT result, GTG and SZ are also suitable areas with grassland vegetation, which is inconsistent with our camera trap data and recent field investigation due to a limited number of camera traps (Figure 2) and investigation area (135 ha) [42]. However, the current potential suitable area of AMD has been taken with caution due to the small areas of XMNNR. Previous studies on AMD have suggested humans [13] and livestock [42] influence their habitat selection. Thus, to understand the influence of seasonal changes and anthropogenic pressures on the activity pattern of the AMD in XMNNR, we investigated seasonal activity patterns and their overlap with anthropogenic disturbances (records of people and livestock). Furthermore, we predicted the current suitable habits of the AMD in the study area to understand their distribution. The study contributes vital information on AMD distribution and resource use for designing effective land sharing and conservation planning in the XMNNR.

## 5. Conclusions

In the XMNNR, we present the first direct evidence of AMD responses to humans and livestock. Based on the results of the study on the identification of the components that influence the choice of habitat by AMD and its distribution, it shows that anthropogenic environmental factors have the greatest impact on AMD, which means that AMD temporally and spatially avoids human activities and livestock grazing to adapt to sympatric distribution. The AMD is present in 75% of camera trap sites and suitable habitats in almost all of the XMNNR except cultivated land and less grassland. The approaches we present are important for comprehending the current drivers of distribution and the amount of suitable habitat in the areas of concern, which may open new opportunities for species, especially for cryptic and elusive behaviors, and for conservation in much wider tracts such as human-dominated landscapes. Therefore, like the XMNNR, a strong monitoring strategy should be adopted in other AMD distribution areas to obtain their distributions for suitable habitat areas prediction or artificial forests planted, and activity patterns to mitigate impacts from livestock and humans.

## Figures and Tables

**Figure 1 animals-12-03061-f001:**
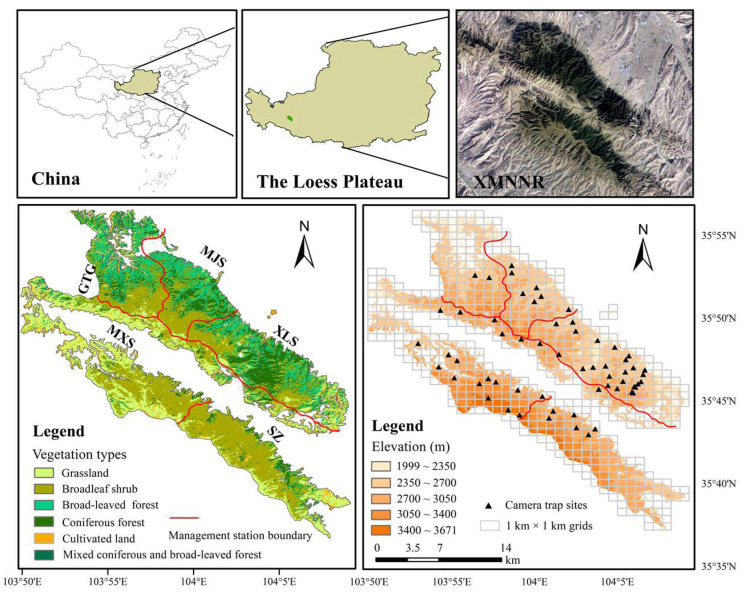
The camera trap sites in the study area and land use types. GTG—Guantangou, MJS—Majiasi, XLS—Xinglongshan, MXS—Maxianshan, and SZ—Shangzhuang.

**Figure 2 animals-12-03061-f002:**
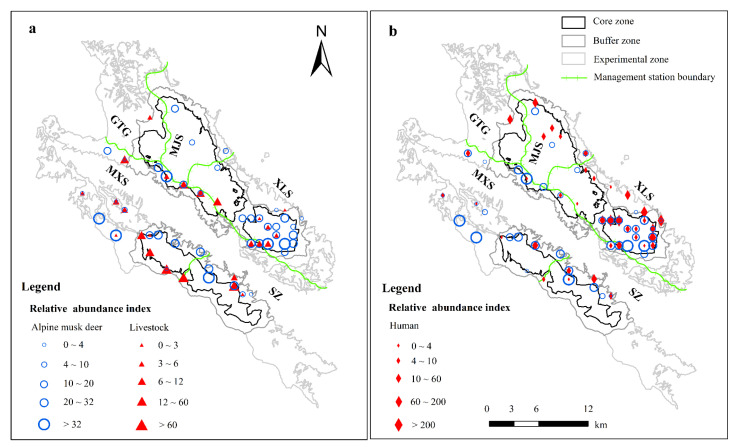
The species captured at coordinate locations in XMNNR (**a**) AMD and livestock (**b**) AMD and human.

**Figure 3 animals-12-03061-f003:**
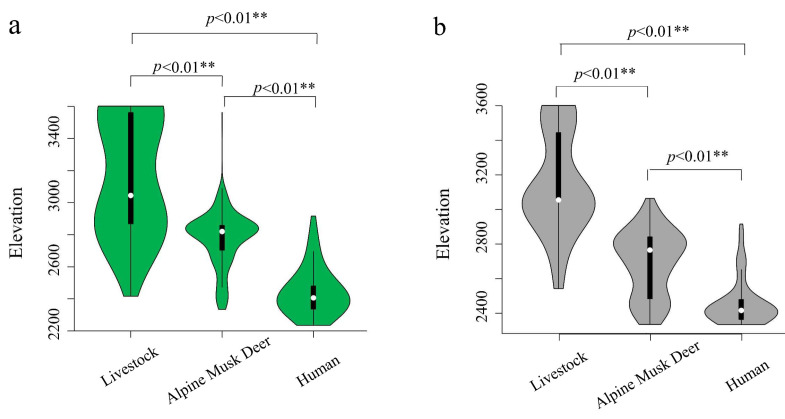
Vertical spatial distribution of livestock, AMD, and human activities during (**a**) green grass period and (**b**) withered grass period.

**Figure 4 animals-12-03061-f004:**
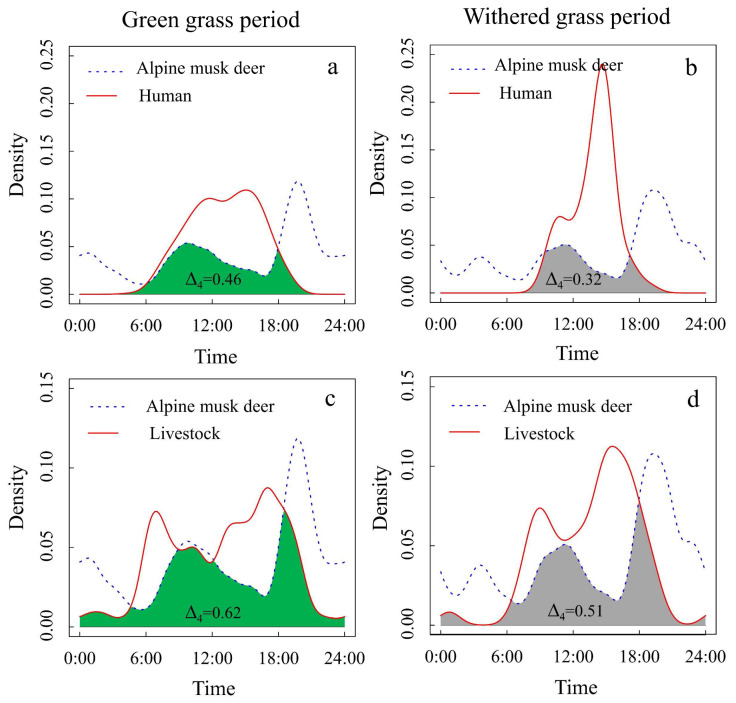
The activity overlap (green area in the green grass period or orange area in the withered grass period) and overlap coefficient Δ of human and livestock activities with AMD. The activity overlaps and overlap coefficient Δ of Alpine musk deer and human activities during (**a**) grass period and (**b**) withered grass period, Alpine musk deer and livestock during (**c**) green grass period and (**d**) withered grass period. (Green areas indicate overlaps during the green grass period or gray areas indicate overlaps during the withered grass period).

**Figure 5 animals-12-03061-f005:**
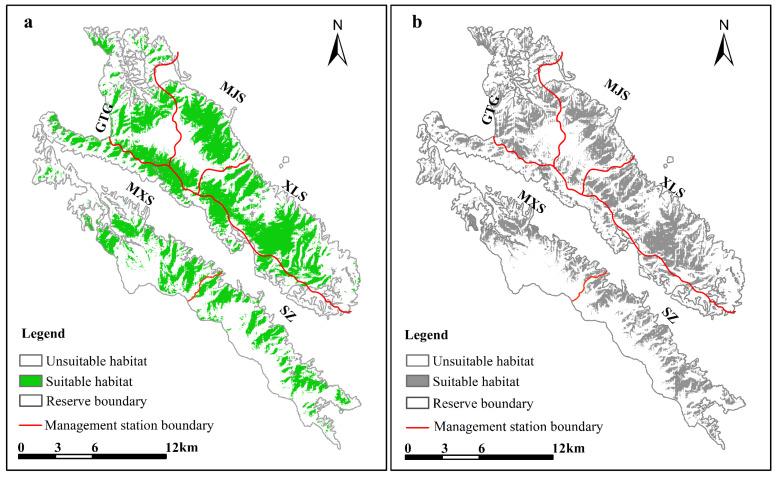
Current climatically suitable area for AMD during two periods as determined by the model: (**a**) green grass period and (**b**) withered grass period.

## Data Availability

The datasets used in this study are available from the corresponding author on reasonable request.

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
