# Peer review of "Alpine Musk Deer (Moschus chrysogaster) Adjusts to a Human-Dominated Semi-Arid Mountain Ecosystem"

_animals, 2022, doi:10.3390/ani12213061_

Round 1
Reviewer 1 Report
It should be noted that the authors undertook a quite interesting study, which should have both application and practical importance in the field of management and protection of an endangered species as a result of multidirectional human activity. Taking action in the field of protection is often of key importance for the survival of many species, especially in areas intensively transformed by man. Therefore, the manuscript submitted for review fits perfectly into the trends of current research in the field of animal species protection.
After reading the manuscript, I had some remarks:
The beginning of the short summary and the summary are identical, this would have to be changed.
Line 154-158, my doubts are raised by the methodical approach, due to the fact that the authors write about a 0.5-hour interval of photos for one or more individuals of a specific species, and how did other animals of the same species appear between these periods?
Are the results surprising in terms of vertical distribution of human activity, Alpine musk deer and domestic animals, regardless of the growing season? What can it be conditioned? Is this thread worth developing?
Line 300, wrong citation?
Line 358-361, recommendation on the protection of the XLS area, the lack of specific recommendations is somewhat unsatisfied, as well as what about the other provinces?
I recommend the article for publication after a slight revision in terms of methodology and editorial requirements, as well as more specific conclusions in the field of protection.
Author Response
We gratefully appreciate you for your carefulness and conscientiousness. Your suggestions are really valuable and helpful for improving our paper. According to your suggestions, we have made the following revisions to this manuscript:
Point 1: The beginning of the short summary and the summary are identical, this would have to be changed.
Response 1: Thank you very much for your helpful comments and constructive suggestions. We have fully modified the simple summary following your suggestions to make it a more inclusive manner for non-science readers to understand.
Point 2: Line 154-158, my doubts are raised by the methodical approach, due to the fact that the authors write about a 0.5-hour interval of photos for one or more individuals of a specific species, and how did other animals of the same species appear between these periods?
Response 2: Thank you for your question. Sorry, it is our mistake. The sentence has been rephrased.
One or more individuals of AMD presenting consecutive photographs taken more than 0.5 h apart and the group size as the maximum number of individuals between 0.5 h due to the difficulty in distinguishing different individuals. The sentence “(ii) consecutive photographs of individuals for different species” has been deleted.
Point 3: Are the results surprising in terms of vertical distribution of human activity, Alpine musk deer and domestic animals, regardless of the growing season? What can it be conditioned? Is this thread worth developing?
Response 3: Thank you for your question. The vertical distribution of human activity, Alpine musk deer, and domestic animals were significantly different despite the growing season, which means they spatially avoid each other during different seasons. Hence, as conservation measures, we might take actions to restrict human and livestock presences to provide enough space for Alpine musk deer.
Point 4: Line 300, wrong citation?
Response 4: Thank you for your reminder. It was the wrong citation and we have changed “Pal et al., 2021” into the number style.
Point 5: Line 358-361, recommendation on the protection of the XLS area, the lack of specific recommendations is somewhat unsatisfied, as well as what about the other provinces?
Response 5: Thank you for your valuable suggestion. We have improved recommendations. See Line 330-333.
We earnestly appreciate your hard work and hope that the correction would meet your approval. Thank you very much for your helpful comments and suggestions.
Reviewer 2 Report
The authors present some very important data on the distribution of Alpine musk deer in relation to cultivated land, distance to the residential area, elevation, aspect, normalized vegetation index, and land cover type. While the results and methods are overall correct and well presented, the authors are missing to highlight to importance of the paper for a broad audience. This is particularly evident in the abstract, but present in introduction and discussion as well. The authors cannot start the summaries with the study site. They need to start framing the theoretical reasoning for the study. The authors cannot have an aim that is so narrow, applied to only one area and one species, if they aim at publishing in a broad journal such as Animals. They need to explain why their study can benefit a broader audience. If they do not solve this issue and make the paper broader (i.e., make comparison with other studies on other species, describe general patterns emerging from their data), next round I would suggest a rejection because the paper would not fit a general journal, but rather an IUCN specialist journal. I also suggest that the authors check their English and read out loud what they wrote as there are many sentences that are not clear, and/or the punctuation is wrong. The authors might need extensive English revision if they do not check carefully what they wrote. There are many repetitions as well, and many sentences are unclear.
Other minor things:
Species names in italics
Check the format of all the numbers, for example 1 019 should be 1,019
The discussion cannot have results of statistical tests, only a quick summary of the results if needed.
Author Response
Response to Reviewer 2 Comments
We gratefully appreciate you for your carefulness and conscientiousness. Your suggestions are really valuable and helpful for improving our paper. According to your suggestions, we have made the following revisions to this manuscript:
Point 1: The authors present some very important data on the distribution of Alpine musk deer in relation to cultivated land, distance to the residential area, elevation, aspect, normalized vegetation index, and land cover type. While the results and methods are overall correct and well presented, the authors are missing to highlight to importance of the paper for a broad audience. This is particularly evident in the abstract, but present in introduction and discussion as well. The authors cannot start the summaries with the study site. They need to start framing the theoretical reasoning for the study. The authors cannot have an aim that is so narrow, applied to only one area and one species, if they aim at publishing in a broad journal such as Animals. They need to explain why their study can benefit a broader audience. If they do not solve this issue and make the paper broader (i.e., make comparison with other studies on other species, describe general patterns emerging from their data), next round I would suggest a rejection because the paper would not fit a general journal, but rather an IUCN specialist journal. I also suggest that the authors check their English and read out loud what they wrote as there are many sentences that are not clear, and/or the punctuation is wrong. The authors might need extensive English revision if they do not check carefully what they wrote. There are many repetitions as well, and many sentences are unclear.
Response: Thanks for your invaluable suggestions. We have complemented importance of the paper for a broad audience in the abstract, introduction, and discussion as well. We have checked our English and read out loud what we wrote as you suggested.
Point 2: The beginning of the short summary and the summary are identical, this would have to be changed.
Response 2: Thank you very much for your helpful comments and constructive suggestions. We have fully modified the simple summary following your suggestions to make it a more inclusive manner for non-science readers to understand.
Point 3: Species names in italics
Response 3: Thank you for your reminder. We have checked all Species names and corrected ‘them into them in italics.
Point 4: Check the format of all the numbers, for example 1 019 should be 1,019
Response 4: Thank you for your reminder. We have checked the format of all the numbers and changed them into right format such 1,019.
Point 5: The discussion cannot have results of statistical tests, only a quick summary of the results if needed.
Response 5: Thank you for your reminder. We have deleted the results of statistical tests and present a quick summary of the results if needed.
Reviewer 3 Report
The paper represents a contribution to the knowledge of the ethology of alpine musk deer and as such is suitable for publication.
Whether AMD is on the game list, whether there are shooting quotas. It is necessary to describe a little better the population of the area, whether the population trends of AMD are positive or negative. Is there predatory pressure on AMD in the study area? Describes human activity in a negative impact on AMD, what does that mean? is this activity related to agriculture, forestry, etc.?
The influence of deforestation and climate change is mentioned, the materials also describe temperature as one of the monitored variables, however there is no presentation of the results of these researches?
Figure 4 .... the activity of wild ruminants in populated areas is usually highest at dawn and dusk so this is not a scientific contribution, it is necessary to better describe the competition between domestic and wild ruminants in the area (competitiveness in food, space, which species of domestic animals appear, etc.)
In the chapter of discussions, the influence of seasonal dynamics in vegetation cover (availability of nutrients) is mentioned as one of the reasons for vertical migration, however in the chapter of results there is no vegetation research that would substantiate such a claim? For such claims, vegetation recordings of a seasonal character and analysis of the stomach or feces are necessary, which would then bring such claims into connection. The paper investigates only the presence of AMD at a certain altitude at a certain time, it is not possible to bring the temperature and vegetation from the results as a reason for migration from the presented results
The paper has grammatical errors, wrong citation and different fonts
Author Response
Response to Reviewer 3 Comments
We gratefully appreciate you for your carefulness and conscientiousness. Your suggestions are really valuable and helpful for improving our paper. According to your suggestions, we have made the following revisions to this manuscript:
Point 1: Whether AMD is on the game list, whether there are shooting quotas. It is necessary to
describe a little better the population of the area, whether the population trends of AMD are
positive or negative. Is there predatory pressure on AMD in the study area? Describes
human activity in a negative impact on AMD, what does that mean? is this activity related
to agriculture, forestry, etc.?
Response 1: Thank you very much for your helpful comments and constructive suggestions. AMD is not on the game list and shooting quotas. We have added the population trends of AMD, and predatory pressure on AMD in the study area, and human activity related to medicinal plants in the introduction. Please see Line
Point 2: The influence of deforestation and climate change is mentioned, the materials also describe
temperature as one of the monitored variables, however, there is no presentation of the
results of these researches?
Response 2: Thank you for your reminder. Temperature as a variable in the XMNNR is stable due to the small area.
Point 3: Figure 4 .... the activity of wild ruminants in populated areas is usually highest at dawn and
dusk so this is not a scientific contribution, it is necessary to better describe the competition
between domestic and wild ruminants in the area (competitiveness in food, space, which
species of domestic animals appear, etc.)
Response 3: Thank you for your suggestion. We have added your suggestion that the activity of AMD in XMNNR is highest at dawn and dusk because of competitiveness in space, which species of domestic animals appear in the discussion.
Point 4: In the chapter of discussions, the influence of seasonal dynamics in vegetation cover
(availability of nutrients) is mentioned as one of the reasons for vertical migration, however
in the chapter of results, there is no vegetation research that would substantiate such a
claim? For such claims, vegetation recordings of a seasonal character and analysis of the
stomach or feces are necessary, which would then bring such claims into connection. The
paper investigates only the presence of AMD at a certain altitude at a certain time, it is not
possible to bring the temperature and vegetation from the results as a reason for migration
from the presented results.
Response 4: Thank you for your reminder. Vegetation recordings of a seasonal character and analysis of the feces have been conducted by Zhang LX but the results are not present for publication. Hence, we only provide a possible explanation.
Point 5: The paper has grammatical errors, wrong citation and different fonts
Response 5: Thank you for your valuable reminder. All grammatical errors, wrong citations, and different fonts are corrected. Please see the whole manuscript.
We earnestly appreciate your hard work and hope that the correction would meet your approval. Thank you very much for your helpful comments and suggestions.
Round 2
Reviewer 2 Report
The authors improved the paper and it is now almost ready to be accepted. I have a few minor comments.
Need to change the species name to italics in the title, as asked before.
The first sentence of the introduction is too long (6 lines), please split into two and add more references.
IUCN categories should be upper case (e.g., Endangered)
The paper is still too focused on AMD management (just now suggested to apply to other areas), it would have been more appropriate to expand the scope of the discussion as I suggested in the previous review. There is a huge literature on anthropogenic disturbance on wildlife populations, and it would have been nice to see this paper fit into the broader literature so that this can be cited more. It would be nice to see, in key parts of the paper, more specific information. For example, in the abstract you just have a list of factors influencing AMD distribution but you do not provide info on how. I know you are limited in space but these are important information in key parts of the paper and can increase your chance of getting cited. I am also wondering if the title can be more specific, like AMD adjusts to a human-dominated Semi-Arid Mountain Ecosystem. Also it should be clearer from the abstract what you mean by adjusts to human activities as it is vague.
Author Response
Point 1: Need to change the species name to italics in the title, as asked before.
Response 1: Thanks for your reminder. We have changed the species name into italics in the title
Point 2: The first sentence of the introduction is too long (6 lines), please split into two and add more references.
Response 2: We have deleted the redundant part of the first sentence of the introduction and added two references.
Point 3: The paper is still too focused on AMD management (just now suggested to apply to other areas), it would have been more appropriate to expand the scope of the discussion as I suggested in the previous review. There is a huge literature on anthropogenic disturbance on wildlife populations, and it would have been nice to see this paper fit into the broader literature so that this can be cited more.
Response 3: We have expanded the scope of the discussion as well as the introduction.
Point 4: It would be nice to see, in key parts of the paper, more specific information. For example, in the abstract, you just have a list of factors influencing AMD distribution but you do not provide info on how.
Response 4: Thanks for your reminder. We have added more detailed information in the abstract, please see lines 31-35.
Point 5: I know you are limited in space but these are important information in key parts of the paper and can increase your chance of getting cited. I am also wondering if the title can be more specific, like AMD adjusts to a human-dominated Semi-Arid Mountain Ecosystem. Also it should be clearer from the abstract what you mean by adjusts to human activities as it is vague.
Response 5: Thanks for your concern. We have changed the title to AMD adjusts to a human-dominated Semi-Arid Mountain Ecosystem.
